# Differences among Sociodemographic Variables, Physical Fitness Levels, and Body Composition with Adherence to Regular Physical Activity in Older Adults from the EXERNET Multicenter Study

**DOI:** 10.3390/ijerph19073853

**Published:** 2022-03-24

**Authors:** Fabio Jiménez-Zazo, Cristina Romero-Blanco, Esther Cabanillas, Asier Mañas, José A. Casajús, Narcís Gusi, Eva Gesteiro, Marcela González-Gross, José-Gerardo Villa-Vicente, Luis Espino-Toron, Ignacio Ara, Susana Aznar

**Affiliations:** 1PAFS Research Group, Faculty of Sports Sciences, University of Castilla-La Mancha, 45071 Toledo, Spain; fabio.jimenez@uclm.es (F.J.-Z.); cristina.romero@uclm.es (C.R.-B.); esther.cabanillas@uclm.es (E.C.); 2Exercise and Health in Special Population Spanish Research Net (EXERNET), 50009 Zaragoza, Spain; asier.manas@uclm.es (A.M.); joseant@unizar.es (J.A.C.); ngusi@unex.es (N.G.); eva.gesteiro@upm.es (E.G.); marcela.gonzalez.gross@upm.es (M.G.-G.); jg.villa@unileon.es (J.-G.V.-V.); luisespinotoron@gmail.com (L.E.-T.); ignacio.ara@uclm.es (I.A.); 3GENUD Toledo Research Group, Faculty of Sports Sciences, University of Castilla-La Mancha, 45071 Toledo, Spain; 4GENUD Research Group, Department of Physiatry and Nursing, Faculty of Health Sciences, University of Zaragoza, 50009 Zaragoza, Spain; 5Centro de Investigación Biomédica en Red de Fisiopatología de la Obesidad y Nutrición (CIBEROBN), 28029 Madrid, Spain; 6International Institute for Aging, 10003 Cáceres, Spain; 7Physical Activity and Quality of Life Research Group (AFYCAV), Faculty of Sport Sciences, University of Extremadura, 10003 Cáceres, Spain; 8CIBER of Frailty and Healthy Aging (CIBERFES), 28029 Madrid, Spain; 9ImFINE Research Group, Department of Health and Human Performance, Faculty of Physical Activity and Sport Sciences-INEF, Universidad Politécnica de Madrid, 28040 Madrid, Spain; 10Grupo de Investigación VALFIS, Instituto de Biomedicina (IBIOMED), Facultad de Ciencias de la Actividad Física y del Deporte, Universidad de León, 24071 León, Spain; 11Unit of Sport Medicine, Cabildo of Gran Canarias, 35019 Las Palmas de Gran Canaria, Spain

**Keywords:** transtheoretical model, stages of change, physical fitness, body composition, sociodemographic variable, older adults

## Abstract

The aim of this study was to explore the differences among between adherence to physical activity (PA) and sociodemographic variables, body composition, and physical fitness levels in older adults (>65 years). A number of 2712 participants (2086 female; 76.92%) ranging from 65 to 92 years, participated in the study. Stages of change (SoC) for PA from the transtheoretical model of change (TTM), together with different sociodemographic variables, physical fitness tests (Senior Fitness Test), and waist and hip circumferences were evaluated. Significant differences were found in age, gender, educational level, current income, physical fitness test, and body composition (all of them, *p* < 0.05), according to the different SoC. Greater adherence to PA practice (action and maintenance stages) was related to better academic level, higher economic income, the male gender, better results in the physical fitness test, and healthier anthropometrics perimeters. Future research is needed to identify the relationship between these variables longitudinally.

## 1. Introduction

The practice of physical activity (PA) has multiple health benefits for older adults [1]. Although there is strong scientific evidence that PA helps to prevent a large number of diseases [2], the levels of non-compliance with recommendations decrease with age, especially from the group of people over 60 years [3].

The World Health Organization (WHO) recommends that older adults perform at least 150–300 min per week of moderate intensity activities or 75–150 min per week of vigorous intensity activities, combined with at least 2 days per week of activities focused on muscle-strengthening and 3 days per week of functional activities [1].

Adequate levels of PA help maintain functional capacity (physical and cognitive) and are related to improvements in physiological systems (metabolic, skeletal, cardiovascular, and immune) and physical fitness levels (cardiorespiratory capacity, flexibility, balance, strength, and power) in older adults [4]. Improvements in physical fitness are related to improvements in activities of daily life (ADL) and thus in the functional capacity and quality of life of older people [5]. Given the importance of PA for the older adults, it is necessary to understand how this behavior becomes a habit [6].

The use of behavioral theories provides us with a context and tools to understand, in depth, the different causes underlying behavioral change (initiation, maintenance, and drop out) for PA. Interventions based on behavioral theories for the promotion of PA have reported increased adherence to this behavior on a regular basis [4,7]. One of the most widely used behavioral models is the transtheoretical model of change (TTM) [7]. This model has proven to be a useful tool in the creation, design, and evaluation of interventions focused on encouraging and promoting the regular practice of PA in the older population [8].

The TTM was proposed by Prochaska and DiClemente [9]; this model assumes that behavioral change is a dynamic process (non-linear), which occurs through a temporal dimension by means of a sequence of stages and processes, through which the individual moves until reaching a regular behavior [9]. The TTM is mainly composed of four constructs: stages of change (SoC), processes of change, decisional balance, and self-efficacy.

The SoC are a descriptive construct, which is a part of the core of this model. SoC explain where people are in their motivation to change and their current behavior change [10]. The SoC categorize individuals from non-intentional to change (precontemplation) to the acquisition of a regular behavior after 6 months (maintenance), and are composed of five stages: Precontemplation, contemplation, preparation, action, and maintenance. [9] and it can help us to measure adherence to PA behavior.

Understanding how adherence to regular PA affects different levels of physical fitness and body composition in the older people could help in the creation, design, and evaluation of tailored intervention programs, with the purpose of maintaining or improving the functional capacity of the older adults. Under this umbrella, the Spanish network of research in exercise and health for special populations (EXERNET) conducted the “EXERNET Multi-center study”. This was the first study measured functional fitness in independent, non-institutionalized Spanish older population [11].

The use of TTM for PA in older healthy adults is scarce [8]. To date, the relationship between adherence to regular PA assessed by Soc and levels of physical fitness, body composition, and sociodemographic variables in the older adults (>65 years) has not been addressed.

Therefore, the aims of this study were: (i) to identify adherence to regular practice of PA through the SoC for PA in a population of non-institutionalized older adults (>65 years), and (ii) to explore the differences among the adherence to this behavior (regular practice of PA) and sociodemographic variables, body composition, and physical fitness level in older adults (>65 years).

## 2. Materials and Methods

This cross-sectional study was carried out from June 2008 to November 2009 on the framework of “EXERNET Multi-center study” [11]. This project involved a representative sample of non-institutionalized Spanish older adults over 65 years. The participants were recruited from 6 regions of Spain: Aragón, Castilla-La Mancha, Castilla y León, Canarias, Extremadura, and Madrid. The exclusion criteria were: people under 65 years; older adults diagnosed with diseases that interfere with their physical performance; those who were living in nursing homes and/or were not independent or able to take care of themselves; those subjects using walking aids. Prior to participation in the trials, all subjects were informed of the objectives of the study, as well as its possible risks and benefits.

3273 older adults were invited to participate in the EXERNET Multi-center study. Finally, a total of 3107 older adults participated in the study, but only 2712 (2086 female; 76.92%) completed the face-to-face interview regarding SoC for PA and met the inclusion criteria, and were therefore included in the present study. Written informed consent was obtained from all participants. The study was performed according to the principles established with the Declaration of Helsinki (1964) as revised in Fortaleza (2013), and approved by the Research Ethics Committees of Aragón (Spain) (14/2008).

All participants were interviewed and evaluated by qualified researchers, previously trained. The information was collected through personal interviews using a structured questionnaire that included current lifestyle habits. It was followed by a physical examination to measure anthropometric variables, and finally the physical fitness tests were carried out.

Sociodemographic characteristics: Personal information, such as age, gender, marital status, living arrangement, education status, and economic status were collected through personal interviews using a structured EXERNET multi-center study questionnaire [11].

Stages of change for physical activity: The SoC for PA behavior were measured through the Physical Activity Stage of Change Questionnaire [12]. This instrument has already been used successfully in older adults to identify their adherence to PA [8]. This tool is composed of 4 dichotomously scaled (yes/no) questions related to regular PA and current intentions.

This tool allows us to categorize participants into 5 stages: precontemplation (inactive, does not think about being more active), contemplation (inactive, but thinks about being more active), preparation (does some PA but is not regular), action (performs PA but for a period of less than 6 months), and maintenance (transform the practice of PA into a habit).

Regular PA or exercise was described as those activities involving brisk walking, running, biking, swimming, or any other activities where the exertion was at least as intense as these activities, at a frequency of at least 30 min/day or more, at least 5 days/week.

Anthropometric Measures: Different perimeters were evaluated using a non-stretchable measuring tape. Waist circumference (WC) was measured with the subject standing upright, feet together, and arms hanging freely at the sides; the measurement was made in midway level between lower edge of the rib cage and iliac crest. Hip circumference (HC) was measured as the maximum circumference around the buttocks posteriorly and the symphysis pubis anteriorly.

Physical Fitness: The “Senior Fitness Test” battery [13] was used in combination with a balance test and a walking speed test, both specific for the older adult population. Static balance by “Flamingo test”: time in seconds that it is able to stand on one foot, up to a maximum of 60-s [14]; lower body strength by the chair stand test: number of times you are able to stand up from a chair in 30 s [13]; upper body strength by arm curl test: number of times you perform a flexo-extension with a dumbbell (2.5 kg for women or 4 kg form men) in 30 s [13]; lower body flexibility by the chair sit-and-reach test: distance in cm from the hands to the tips of the feet when flexing the trunk while sitting [13]; upper body flexibility by back scratch test: distance in centimeters between both hands when trying to touch each other diagonally from behind [13]; agility/dynamic balance by the 8 foot up-and-go test: time in seconds it takes to make a circuit starting from seating [13]; speed by the 30 m walk by brisk walking test: time in seconds it takes to walk 30 m [15]; and aerobic endurance by the 6 min walk test: meters you are able to walk in 6 min [13].

All the tests were performed only once, except the one leg test (balance), which was performed twice with each leg, as well as the 8 foot up-and-go test (agility/dynamic balance) and the 30 m walk test (speed), which were also performed twice.

Statistical analysis: Descriptive statistics (reported as mean and SD, number of participants, and percentage, according to the nature of these variables) and chi-square test were used to summarize and compare the sociodemographic variables regarding to the SoC. Tests for normality were performed by implementing Kolmogorov-Smirnov tests and parametric statistisl (*p* > 0.05) were applied. ANOVA tests were used to determine significant differences between the anthropometrics variables and physical fitness test for each SoC and the Scheffé post-hoc test was applied to determinate differences between groups. Statistics SPSS v25.0 software was used to analyze the data (SPSS Inc., Chicago, IL, USA). The level of significance was set at *p* < 0.05.

## 3. Results

The sample comprised a total of 2712 participants (2086 female; 76.92%) ranging from 65 to 92 years (72.32 ± 5.32 years). According to the individual SoC, most participants were in the maintenance stage (*n* = 1801; 66.4%), followed by preparation stage (*n* = 536; 19.8%), and then precontemplation stage (*n* = 166; 6.1%), contemplation stage (*n* = 108; 4.0%), and action stage (*n* = 101; 3.7%). The sociodemographic characteristics are shown in Table 1.

We found significant differences between the sociodemographic variables of age (*p* < 0.001), gender (*p* < 0.001), educational level (*p* < 0.001), and current income (*p* = 0.001) according to the different SoC. Table 2 shows the means and standard deviations of each of the physical fitness tests and body composition perimeters in accordance with each of the SoC. It can be observed how, as we moved up through the SoC, i.e., to a more physically active SoC, there was an improvement in the values of physical fitness and body composition.

In relation to the physical fitness variables, the results obtained in the balance test, strength test (legs and arms), flexibility test (legs and arms), and endurance test improved from the earliest to the most advanced stages (*p* < 0.001). According to the speed and agility/dynamic balance tests, those who were in the initial SoC (precontemplation, contemplation, and preparation stages) took more time to perform the tests and therefore scored worse results than older adults in maintenance SoC (*p* < 0.001).

Regarding waist and hip perimeters, those older adults who were in advanced SoC (action and maintenance stages) registered healthier perimeters.

According to the Scheffé post-hoc tests, older adults in the maintenance stage had better scores/results in both physical fitness and body composition that those in lower stages (precontemplation, contemplation, or preparation stages).

## 4. Discussion

The purpose of this study was to identify the differences among the sociodemographic variables, body composition, and physical fitness level with current adherence to PA through the use of SoC for regular PA, in Spanish older adults over 65 years.

Understanding the determinants of PA behavior is essential for the design, creation, and implementation of future physical exercise programs aimed at improving the health of older people. The use of TTM for PA in older healthy adults is scarce [8]. To date, the relationship between adherence to regular PA assessed by Soc and levels of physical fitness, body composition, and sociodemographic variables in the older adults (>65 years) has not been addressed.

According to the sociodemographic variables, our results have identified that those older people with greater adherence to PA practice (classified into action and maintenance stages, i.e., older adults with less than 6 months of regular practice of PA and those with who have achieved a quite permanent PA practice) have a better academic level, higher economic income, and are male, compared to older adults in the initial SoC (older people classified into precontemplation stage, i.e., older adults not interested in PA practice and those considering being physically active). Globally, it has been observed that PA practice decreases with age and the proportion of women who drop out this behavior over time is higher than that of men [3,16]. In Spain, according to the latest data collected by the National Health Survey [17], the same trend was identified for gender and age in terms of PA practice dropout; only 9.2% of the population over 65 years practiced regular PA in their leisure time [18]. On the other hand, low socioeconomical status (low educational levels and low economic income) are related to low levels of PA [19,20], in addition this status are related with health problems such as an increased cardiovascular disease risk [21], and overweight and obesity problems [22] in adulthood. In Spain, less favored social classes had longer periods of sedentary lifestyles and greater non-compliance with PA recommendations [17].

In relation to physical fitness, older adults classified in advanced SoC recorded better results in all tests (upper and lower extremity strength and flexibility, balance, gait speed, and endurance) when compared to older people in lower SoC. Low physical fitness levels leads to a decrease in muscle function, affecting physical function and gait speed [23,24,25], and are considered as predicting factor of a decline in the ability to perform the activities of daily living [26]. Therefore, identifying older adults with low adherence to PA is critical key because they are at risk of developing a deterioration in their ability to perform activities of daily living; this is essential to prevent more severe limitations (loss of independence and disability status) in the older adult’s daily functioning.

Previous studies in older adults have shown how TTM-based interventions help to improve their physical fitness levels and achieve a progression towards more proactive SoC (action and maintenance stages) [27,28,29]. Therefore, it seems a good recommendation to evaluate the initial SoC for PA in older adults before the design of an intervention, and hence highlights the purpose of the future applications of this study.

The results obtained also showed a healthy relationship between anthropometric measurements (waist and hip perimeters), body composition, and the SoC. High waist and hip perimeter are related to an increase of the risk of cardiovascular diseases [30], different pathologies [31], and an increase the risk of becoming overweight or obese [31]. Overweightness and obesity are related to low PA levels [32] and longer periods of sedentary lifestyles [33]. Sedentary behaviors are associated with increased risk of all-cause mortality [34]. This vicious circle could limit the practice of PA in older adults. Encouraging the promotion of people towards more advanced stages will favor the health of older adults.

Our results seem to confirm that sedentary older adults with no intention to be physically active and those who remain sedentary while considering being active did not have good physical fitness and waist and hip perimeters. On the contrary, older adults who are regularly active had better physical fitness and healthy waist and hip perimeters. Therefore, the SoC for PA confirms the effect of regular PA, i.e., improving physical fitness and waist and hip perimeters as we become more physically active.

This study supports the use of the SoC in older adults and its relationship with fitness levels. The use of SoC has identified more advanced SoC (action and maintenance) with better fitness levels and the reserve for lower SoC. These results promote the use of interventions tailored to each SoC using the TTM constructs (processes of change, decisional balance (benefits and barriers), and self-efficacy) to achieve a more successful behavioral change [35] and to increase fitness and health levels in older adults.

The present study has several strengths: the sample size (*n* = 2712), the age of the participants (all of them >65 years), and a representative sample of Spanish older adults. However, we have some limitations. According to the PA behavior, although the questionnaire for the assessment the adherence for regular PA has been previously validated [12], objective measures of PA were not included. Even though the sample is a nationally representative sample, it only includes independent, non-institutionalized older adults aged between 65–92 years, generally from sport and civic centers, so the results must be interpreted with caution, since there may be a bias for the extrapolation of these results to other groups of older population. This is a cross-sectional study and therefore we can only identify the associations between SoC for PA and physical fitness and anthropometric values. Future research is needed to identify the relationship between these variables in different population groups of older adults, with a more specific age distribution (due to the great heterogeneity of this population), as well as research over the long term, with a longitudinal study design that identifies the evolution of behavioral change in older adults and the relationship with different sociodemographic variables, physical fitness levels, body composition, or other variables related with adherence to regular PA.

We can conclude that our research supports that the SoC is a useful tool to measure adherence to PA behavior in older adults. Moreover, it confirms that those older adults in more advanced SoC have also better fitness levels and healthier anthropometric measurements. Therefore, in the light of our results, we promote the use of SoC in further studies focused on PA behavior.

## 5. Conclusions

This study examined the SoC for PA behavior among older adults according to sociodemographic variables, body anthropometrics measurements, and physical fitness level. Physical fitness and body anthropometrics measurement scores obtained from these variables differed according to the SoC. Greater adherence to PA practice (action and maintenance stages) were related to better academic level, higher economic income, the male gender, better results in physical fitness test, and healthier anthropometrics perimeters. Moreover, the SoC for PA confirms the effect of regular PA, i.e., improving physical fitness and waist and hip perimeters as we become more physically active. The results obtained from this study will allow the creation and development of future tailored PA interventions. Future research is needed to identify the relationship between these variables over the long term.

## Figures and Tables

**Table 1 ijerph-19-03853-t001:** Socio-demographic characteristics and stages of change for physical activity. Bivariate Analysis.

		Total Sample n2712	Stages of Change					
			PC n166	C n108	Pp n536	A n101	Mt n1801	Sig
Age		72.3 ± 5.3	73.8 ± 6.2 *†^β^	71.30 ± 5.1	72.6 ± 5.1	71.1 ± 5.1	72.2 ± 5.2	<0.001
Gender	Man	626 (23.1%)	74 (11.8%)	32 (5.1%)	85 (13.6%)	21 (3.4%)	414 (66.1%)	<0.001
	Woman	2086 (76.9%)	92 (4.4%)	76 (3.6%)	451 (21.6%)	80 (3.8%)	1387 (51.1%)	
Marital Status	Single	131 (4.9%)	5 (3.8%)	2 (1.5%)	24 (18.3%)	8 (6.1%)	92 (70.2%)	0.261
	Married	1647 (61.9%)	93 (5.6%)	67 (4.1%)	339 (20.6%)	59 (3.6%)	1089 (66.1%)	
	Divorced	72 (2.7%)	4 (5.6%)	6 (8.3%)	11 (15.3%)	3 (4.2%)	48 (66.7%)	
	Widowed	812 (30.5%)	60 (7.4%)	33 (4.1%)	158 (19.5%)	23 (2.8%)	538 (66.3%)	
Living alone	No	1960 (73.2%)	118 (6.0%)	76 (3.9%)	395 (20.2%)	72 (3.7%)	1299 (66.3%)	0.936
	Yes	716 (26.8%)	46 (6.4%)	31 (4.3%)	138 (19.3%)	29 (4.1%)	472 (65.9%)	
Academic Level	Cannot read or write	289 (10.8%)	25 (8.7%)	15 (5.2%)	82 (28.4%)	7 (2.4%)	160 (55.4%)	<0.001
	Primary Studies	1915 (71.7%)	118 (6.2%)	66 (3.4%)	386 (20.2%)	71 (3.7%)	1274 (66.5%)	
	Hight school Studies	316 (11.8%)	16 (5.1%)	19 (6.0%)	40 (12.7%)	14 (4.4%)	227 (71.8%)	
	University Studies	151 (5.7%)	6 (4.0%)	7 (4.6%)	23 (15.2%)	9 (6.0%)	106 (70.2%)	
Current Income	−600 €/month	1029 (42.4%)	67 (6.5%)	42 (4.1%)	249 (24.2%)	42 (4.1%)	629 (61.1%)	0.001
	600–900 €/month	774 (31.9%)	52 (6.7%)	31 (4.0%)	130 (16.8%)	28 (3.6%)	533 (68.9%)	
	+900 €/month	622 (25.6%)	33 (5.3%)	27 (4.3%)	98 (15.8%)	26 (4.2%)	438 (70.4%)	

M: Mean; SD: Standard Deviation; PC: Precontemplation & Contemplation stages; Pp: Preparation stage; AM: Action & Maintenance stages; * PC > C; † PC > A; ^β^ PC > Mt; Statistical significance: *p* < 0.05.

**Table 2 ijerph-19-03853-t002:** Body composition, physical fitness level, and stages of change for physical activity. Bivariate analysis.

	Stages of Change						
	PC	C	Pp	A	Mt	Sig	Post-Hoc
Physical Fitness (Senior Fitness Test)							
Balance (s)	21.9 ± 20.1	22.7 ± 19.8	22.4 ± 19.8	24.2 ± 20.1	28.9 ± 21.6	<0.001	Mt > PC *; Mt > Pp **
Lower body strength (rep)	13.5 ± 3.9	13.4 ± 3.1	13.8 ± 3.3	14.0 ± 3.6	14.8 ± 3.5	<0.001	Mt > PC **; Mt > C *; Mt > Pp **
Right upper strength (rep)	15.0 ± 3.9	15.4 ± 3.7	15.5 ± 3.8	16.0 ± 3.8	16.8 ± 3.8	<0.001	Mt > PC **; Mt > C *; Mt > Pp **
Left upper strength (rep)	15.4 ± 3.9	15.8 ± 3.8	15.5 ± 3.9	16.0 ± 4.2	16.9 ± 4.0	<0.001	Mt > PC **; Mt > Pp **
Right upper flexibility (cm)	−15.4 ± 13.5	−12.1 ± 11.9	−10.9 ± 12.3	−11.3 ± 14.5	−8.8 ± 11.2	<0.001	PC > Pp *; PC > Mt **; Pp > Mt *
Left upper flexibility (cm)	−19.3 ± 11.4	−17.1 ± 10.8	−15.4 ± 12.3	−15.6 ± 13.8	−13.2 ± 10.7	<0.001	PC > Pp *; PC > Mt **; C > Mt *; Pp > Mt *
Right lower flexibility (cm)	−6.7 ± 10.2	−5.4 ± 10.7	−5.5 ± 10.8	−3.3 ± 10.0	−3.1 ± 10.7	<0.001	PC > Mt *; Pp > Mt *
Left lower flexibility (cm)	−6.6 ± 10.8	−5.4 ± 10.6	−4.8 ± 10.7	−3.4 ± 10.3	−2.9 ± 10.7	<0.001	PC > Mt *; Pp > Mt *
Agility/dynamic balance (s)	6.5 ± 2.2	6.3 ± 1.8	6.2 ± 1.7	6.2 ± 1.9	5.7 ± 1.4	<0.001	PC > Mt **; C > Mt *; Pp > Mt **
30-m Walk test (s)	18.7 ± 5.6	18.1 ± 4.5	18.0 ± 3.5	18.6 ± 4.9	16.9 ± 3.5	<0.001	PC > Mt **; Pp > Mt **; A > Mt *
Aerobic endurance (m)	483.6 ± 98.4	492.1 ± 85.9	505.9 ± 84.1	506.5 ± 107.2	537.9 ± 87.7	<0.001	Mt > PC **; Mt > C **; Mt > Pp **; Mt > A *
Anthropometrics Variables							
Waist circumference (cm)	98.1 ± 13.0	95.1 ± 11.6	95.7 ± 12.3	94.9 ± 13.5	93.1 ± 11.8	<0.001	PC > Mt **; Pp > Mt *
Hip circumference (cm)	105.0 ± 9.1	105.7 ± 7.0	106.4 ± 9.4	106.5 ± 9.4	103.9 ± 8.9	<0.001	Pp > Mt **

M: Mean; SD: Standard Deviation; PC: Precontemplation & Contemplation stages; Pp: Preparation stage; AM: Action & Maintenance stages; * Statistical significance: *p* < 0.05; ** Statistical significance: *p* < 0.001.

## Data Availability

Not applicable.

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
