# Peer review of "Differences among Sociodemographic Variables, Physical Fitness Levels, and Body Composition with Adherence to Regular Physical Activity in Older Adults from the EXERNET Multicenter Study"

_ijerph, 2022, doi:10.3390/ijerph19073853_

Round 1
Reviewer 1 Report
The authors performed an interesting study on the large cohort consisting of 2.712 both older and very old adults (>85 years). The authors were successful in showing the significant differences between various stages of change (SoC) for physical activity (PA) and factors like age, gender, educational level, current income, physical fitness test, and body composition. What should be emphasized, up to date, the relationship between adherence to regular PA assessed by Soc for PA and levels of physical fitness, body composition and sociodemographic variables in the older adults (>65 years) has not been yet addressed by any other researchers.
Minor remarks:
The text should be checked to correct the tiny editorial mistakes like, for example, those given below:
Lines 81-83: Comas should be rearranged in the sentence to make it understandable: "Under this umbrella, the Spanish network of research in exercise and health for special populations (EXERNET), conducted the “EXERNET Multi-center study”,. This was the first study measured functional fitness in independent non-institutionalized Spanish older population.
Line 85: Soc for PA - should be "SoC".
Line 110: "The information was collected through personal interviews using a structured questionnaire that included current lifestyle habits, followed by a physical examination to measure anthropometric variables and finally the physical fitness tests were carried out." Should be restructured, e.g.: "The information was collected through personal interviews using a structured questionnaire that included current lifestyle habits. It was followed by a physical examination to measure anthropometric variables, and finally the physical fitness tests were carried out."
Line 232: "The results obtained also showed a healthy relationship between anthropometric measurements (waist and hip perimeters) (coma needed) body composition and the SoC.
Author Response
We thank the editor and reviewers for their time and effort to improve our manuscript. We will address each comment individually below. “R” will be used for the reviewer’s comments and “A” will be used for the authors’ answers.
R1: Lines 81-83: Comas should be rearranged in the sentence to make it understandable: "Under this umbrella, the Spanish network of research in exercise and health for special populations (EXERNET), conducted the “EXERNET Multi-center study”,. This was the first study measured functional fitness in independent non-institutionalized Spanish older population.
A: Corrected
R1: Line 85: Soc for PA - should be "SoC".
A: Corrected
R1: Line 110: "The information was collected through personal interviews using a structured questionnaire that included current lifestyle habits, followed by a physical examination to measure anthropometric variables and finally the physical fitness tests were carried out." Should be restructured, e.g.: "The information was collected through personal interviews using a structured questionnaire that included current lifestyle habits. It was followed by a physical examination to measure anthropometric variables, and finally the physical fitness tests were carried out."
A: Corrected
R1: Line 232: "The results obtained also showed a healthy relationship between anthropometric measurements (waist and hip perimeters) (coma needed) body composition and the SoC.
A: Corrected
Reviewer 2 Report
The paper is interesting and could be a welcome addition to the research elderly field. However, several suggestions need to be addressed before the present paper can be considered for publication.
Please write what is the primary and secondary outcomes. Please make aim clearer.
Please delete the A from first line in Discussion
Add limitations of study and add more details on future research.
Please add the following new REF:
Effects of multicomponent exercise training intervention on hemodynamic and physical function in older residents of long-term care facilities: A multicenter randomized clinical. Journal of Bodywork and Movement Therapies 28, 231-237
https://doi.org/10.1016/j.jbmt.2021.07.009
Pepera G, Krinta K, Mpea C, Antoniou V, Peristeropoulos A, Dimitriadis Z. Randomized Controlled Trial of Group Exercise Intervention for Fall Risk factors Reduction in nursing home residents. Canadian Journal on Aging, 42 (1) (in press).
Gallardo-Alfaro, L.; Bibiloni, M.d.M.; Mateos, D.; Ugarriza, L.; Tur, J.A. Leisure-Time Physical Activity and Metabolic Syndrome in Older Adults. Int. J. Environ. Res. Public Health 2019, 16, 3358. https://doi.org/10.3390/ijerph16183358
Marcos-Pardo, P.J.; González-Gálvez, N.; López-Vivancos, A.; Espeso-García, A.; Martínez-Aranda, L.M.; Gea-García, G.M.; Orquín-Castrillón, F.J.; Carbonell-Baeza, A.; Jiménez-García, J.D.; Velázquez-Díaz, D.; Cadenas-Sanchez, C.; Isidori, E.; Fossati, C.; Pigozzi, F.; Rum, L.; Norton, C.; Tierney, A.; Äbelkalns, I.; Klempere-Sipjagina, A.; Porozovs, J.; Hannola, H.; Niemisalo, N.; Hokka, L.; Jiménez-Pavón, D.; Vaquero-Cristóbal, R. Sarcopenia, Diet, Physical Activity and Obesity in European Middle-Aged and Older Adults: The LifeAge Study. Nutrients 2021, 13, 8. https://doi.org/10.3390/nu13010008
Author Response
We thank the editor and reviewers for their time and effort to improve our manuscript. We will address each comment individually below. “R” will be used for the reviewer’s comments and “A” will be used for the authors’ answers.
R2: Please write what is the primary and secondary outcomes. Please make aim clearer.
A: Corrected
R2: Please delete the A from first line in Discussion
A: Corrected
R2: Add limitations of study and add more details on future research.
A: A new paragraph has been included in the discussion in regards of the limitations and future research.
R2: Please add the following new REF:
A: Included
Reviewer 3 Report
The study claims addressing the relationship between sociodemographic variables, body composition and physical fitness level with adherence to regular physical activity (PA) in older adults. Adherence to PA is indicated by the participants’ stage according to the Transtheoretical Model (TTM). TTM is one of the most important contemporary behavioural models in understanding the processes during the transition from an unhealthy to a healthy behaviour. Taking also into account the large sample size, the study is quite important in the research of physical activity in aging. There are some points however, that should be reconsidered before the manuscript merit publication.
A main point that should be considered is the wording in the title and the aim of the study. Examining a “relationship” points to comparing at least two continuous variables, which is not the case in the current study. TTM model provides a taxonomy of 5 stages, according to the progress of change. According to the design of the study, the results outcome is differences among people at the various stages of the TTM model. These stages however, are different conditions and not necessarily a continuum in behaviour change. For example, someone, which is currently in the pre-contemplation stage, might not ever reach even the preparation stage. I am suggesting that the title of the manuscript and the description of the aim of the study should be “differences among …” and not “relationship between …”.
The arguments in the discussion section should be based on differences among people at the different stages and not on relationships. You should concentrate to the specific processes in each stage when discussing the different characteristics of people in these stages, taking into account that people at different stage have achieved a different goal. For example, people at the “maintenance stage” have achieved a quite permanent change, while people at the “action stage” are trying to change their behaviour and eventually they might succeed or not. On the contrary, people at the pre-contemplation might simply not interested in changing their behaviour at all. Discussing the above differences may lead to more fruitful and sound conclusions.
One more point that could improve the quality in the presentation of the study is statistics. The statistical analysis used seems to be appropriate but you should report the assumptions in using this analysis. For example you should report the indexes about the data normal distribution (skewness and kurtosis), or homogeneity index, since there are very large number differences among groups.
The study claims addressing the relationship between sociodemographic variables, body composition and physical fitness level with adherence to regular physical activity (PA) in older adults. Adherence to PA is indicated by the participants’ stage according to the Transtheoretical Model (TTM). TTM is one of the most important contemporary behavioural models in understanding the processes during the transition from an unhealthy to a healthy behaviour. Taking also into account the large sample size, the study is quite important in the research of physical activity in aging. There are some points however, that should be reconsidered before the manuscript merit publication.
A main point that should be considered is the wording in the title and the aim of the study. Examining a “relationship” points to comparing at least two continuous variables, which is not the case in the current study. TTM model provides a taxonomy of 5 stages, according to the progress of change. According to the design of the study, the results outcome is differences among people at the various stages of the TTM model. These stages however, are different conditions and not necessarily a continuum in behaviour change. For example, someone, which is currently in the pre-contemplation stage, might not ever reach even the preparation stage. I am suggesting that the title of the manuscript and the description of the aim of the study should be “differences among …” and not “relationship between …”.
The arguments in the discussion section should be based on differences among people at the different stages and not on relationships. You should concentrate to the specific processes in each stage when discussing the different characteristics of people in these stages, taking into account that people at different stage have achieved a different goal. For example, people at the “maintenance stage” have achieved a quite permanent change, while people at the “action stage” are trying to change their behaviour and eventually they might succeed or not. On the contrary, people at the pre-contemplation might simply not interested in changing their behaviour at all. Discussing the above differences may lead to more fruitful and sound conclusions.
One more point that could improve the quality in the presentation of the study is statistics. The statistical analysis used seems to be appropriate but you should report the assumptions in using this analysis. For example you should report the indexes about the data normal distribution (skewness and kurtosis), or homogeneity index, since there are very large number differences among groups.
Author Response
We thank the editor and reviewers for their time and effort to improve our manuscript. We will address each comment individually below. “R” will be used for the reviewer’s comments and “A” will be used for the authors’ answers.
R3: A main point that should be considered is the wording in the title and the aim of the study. Examining a “relationship” points to comparing at least two continuous variables, which is not the case in the current study. TTM model provides a taxonomy of 5 stages, according to the progress of change. According to the design of the study, the results outcome is differences among people at the various stages of the TTM model. These stages however, are different conditions and not necessarily a continuum in behaviour change. For example, someone, which is currently in the pre-contemplation stage, might not ever reach even the preparation stage. I am suggesting that the title of the manuscript and the description of the aim of the study should be “differences among …” and not “relationship between …”.
A: Thanks for this relevant comment, we have changed it.
R3: The arguments in the discussion section should be based on differences among people at the different stages and not on relationships. You should concentrate to the specific processes in each stage when discussing the different characteristics of people in these stages, taking into account that people at different stage have achieved a different goal. For example, people at the “maintenance stage” have achieved a quite permanent change, while people at the “action stage” are trying to change their behaviour and eventually they might succeed or not. On the contrary, people at the pre-contemplation might simply not interested in changing their behaviour at all. Discussing the above differences may lead to more fruitful and sound conclusions.
A: Two new paragraphs at the discussion have been included. At the conclusion a new sentence has also been included.
R3: One more point that could improve the quality in the presentation of the study is statistics. The statistical analysis used seems to be appropriate but you should report the assumptions in using this analysis. For example you should report the indexes about the data normal distribution (skewness and kurtosis), or homogeneity index, since there are very large number differences among groups.
A: A new sentence has been included in the methodology section addressing this issue.
Reviewer 4 Report
Interesting article about the relationship between sociodemographic variables, physical fitness levels and body composition with adherence to regular physical activity
Physical activity is considered to be one of the major constituents of a healthy lifestyle. It plays a key role in the promotion of health and in prophylaxis, contributing to a lower incidence of diseases.
Below are presented detailed comments, which according to the reviewer, should be included in the manuscript or should be responded to by the Authors:
Materials and Methods
- large age distribution was used: 65 – 92 years – it seems so large age distribution can influence the conclusion? Have the authors included this in the research planning phase? It is clear that age has an impact on the level of physical activity and physical fitness and in the exclusion criteria was included only age below 65 years
- the exclusion criteria include only cancer and dementia. What is the reason for this choice? Other medical illnesses may also influence activity and fitness levels
Results:
- line: 177-179 – “We found significant differences between the sociodemographic variables of age, gender, educational level and current income according to the different SoC (p<0.001; p=0.001)” - I propose to determine the level of significance for each variable separately
- table 2 - I propose to add to post hoc significance level.
I recommend publishing the manuscript after prior correction of the text or the Authors’ reference to the specific comments
Author Response
We thank the editor and reviewers for their time and effort to improve our manuscript. We will address each comment individually below. “R” will be used for the reviewer’s comments and “A” will be used for the authors’ answers.
R4: Large age distribution was used: 65 – 92 years – it seems so large age distribution can influence the conclusion? Have the authors included this in the research planning phase? It is clear that age has an impact on the level of physical activity and physical fitness and in the exclusion criteria was included only age below 65 years
A: we chose 65 years and plus, as an inclusion criteria due to in Spain it is the age when older adults normally stop their jobs and it is a big change in their lifestyles. We include all older who did not have any disease which could stop them from being independent. Our results reported that age was a factor which was related to the stages of change as expected.
R4: The exclusion criteria include only cancer and dementia. What is the reason for this choice? Other medical illnesses may also influence activity and fitness levels
A: The reviewer is right; we did not include older adults with illnesses that interfere with their physical performance. Cancer and dementia were chosen in the case they were diagnosed with those diseases. We have changed the text accordingly: “older adults diagnosed with diseases that interfere with their physical performance”
R4: Line: 177-179 – “We found significant differences between the sociodemographic variables of age, gender, educational level and current income according to the different SoC (p<0.001; p=0.001)” - I propose to determine the level of significance for each variable separately
A: It has been included.
R4: Table 2 - I propose to add to post hoc significance level.
A:it has been added.